# HAINAN: Fast and Accurate Transducer for Hybrid-Autoregressive ASR

**Hainan Xu, Travis M. Bartley, Vladimir Bataev, Boris Ginsburg**
NVIDIA Corp., USA
{hainanx,tbartley,vbataev}@nvidia.com

## Abstract

We present **H**ybrid-**A**utoregressive **IN**ference TrANsducers (HAINAN), a novel architecture for speech recognition that extends the Token-and-Duration Transducer (TDT) model. Trained with randomly masked predictor network outputs, HAINAN supports both autoregressive inference with all network components and non-autoregressive inference without the predictor. Additionally, we propose a novel semi-autoregressive inference method that first generates an initial hypothesis using non-autoregressive inference, followed by refinement steps where each token prediction is regenerated using parallelized autoregression on the initial hypothesis. Experiments on multiple datasets across different languages demonstrate that HAINAN achieves efficiency parity with CTC in non-autoregressive mode and with TDT in autoregressive mode. In terms of accuracy, autoregressive HAINAN achieves parity with TDT and RNN-T, while non-autoregressive HAINAN significantly outperforms CTC. Semi-autoregressive inference further enhances the model's accuracy with minimal computational overhead, and even outperforms TDT results in some cases. These results highlight HAINAN's flexibility in balancing accuracy and speed, positioning it as a strong candidate for real-world speech recognition applications.

## 1 Introduction

End-to-end neural automatic speech recognition (ASR) has seen significant advancements in recent years, namely due to the development of three architecture paradigms: Connectionist Temporal Classification (CTC) (Graves et al., 2006), Recurrent Neural Network Transducers (RNN-T) (Graves, 2012), and Attention-based Encoder and Decoder Models (Chorowski et al., 2015; Chan et al., 2016). These models have gained widespread adoption, supported by open-source projects such as ESPNet (Watanabe et al., 2018), SpeechBrain (Ravanelli et al., 2021), and NeMo (Kuchaiev et al., 2019), etc. Of those models, CTC and RNN-T share a frame-synchronous design, enabling streaming processing of speech input. However, they differ in their inference approaches: CTC models support non-autoregressive (NAR) inference based on a conditional-independence assumption, while RNN-T models require autoregressive (AR) inference due to their dependence on the history context of partial hypotheses. This fundamental difference creates a trade-off between accuracy and efficiency, with CTC models offering greater speed but typically lower accuracy than Transducers.

Our work bridges this gap with ***H**ybrid-**A**utoregressive **IN**ference TrANsducer* (HAINAN)[1]: a novel implementation of RNN-T capable of non-autoregressive, autoregressive and, *semi-autoregressive* inference. Our contributions are as follows:

- We propose a method to train a Token-and-Duration Transducer (TDT) model by stochastically masking out the predictor. The approach is easy to implement and only requires a one-line change in the training code of token-and-duration model implementation.

- We present a novel *semi-autoregressive* (SAR) inference method that first uses NAR to generate initial hypotheses, and then uses autoregression to refine the hypotheses in parallel iteratively.

---

[1]Despite similar names, our work is unrelated to *Hybrid Autoregressive Transducer* (Variani et al., 2020).

- We present a means to model the output of the NAR HAINAN model containing the token and duration predictions as a *directed acyclic graph* (DAG), and propose a simple Viterbi-based decoding method for inference, with further accuracy gains.

Our results demonstrate the following:

- AR HAINAN achieves on-par WER accuracy with RNN-T and TDT models, with inference speeds similar to TDT-based decoding, and significantly faster than RNN-T inference.
- NAR HAINAN outperforms CTC accuracy while maintaining similar efficiency. Our investigation shows that NAR HAINAN can better capture potential ambiguities in the speech by learning more nuanced representations, a capability that frame-by-frame NAR like CTC inherently lacks.
- SAR consistently brings accuracy gains over NAR, approaching the performance of AR inference with minimal computational overhead.
- HAINAN's high performance is due to its learning of more diverse encoder outputs, a property that allows our Viterbi decoder to achieve the same accuracy of AR inference *despite the use of non-autoregressive outputs*.

We will open-source our implementation and release trained model checkpoints for public use.

## 2 RELATED WORK

Recent research in speech recognition has focused on enhancing WER accuracy while mitigating reductions in inference speed. This has been most notable for self-attention-based architectures, with improvements in latency for Transformer models (Yeh et al., 2019) leading to creation of the Conformer (Gulati et al., 2020) and it variants (e.g. Zipformer (Yao et al., 2023), FastConformer (Rekesh et al., 2023)). While all have notable architectural improvements, they can be generalized as attempts to improve model capacity and/or reducing the latency caused by attention-based inference over large audio inputs.

For Transducer models specifically, various optimizations have been explored. He et al. (2019) proposed a caching mechanism for the RNN-T predictor to reduce redundancy in inference computations. Jain et al. (2019) introduced a pruning method for RNN-T beam search to reduce the search space, resulting in faster decoding. Ghodsi et al. (2020) investigated predictors with stateless networks for Transducer models. While this approach reduced the computational complexity of the original LSTM predictor, it came at the cost of slight accuracy degradation. Kang et al. (2023) proposed monotonic constraints on RNN-T training and inference, reducing the number of steps RNN-T inference stays on the same frame, and reported faster inference with those models. Bataev et al. (2024) proposed a novel label-looping decoding algorithm for Transducers, using parallel GPU calls for the majority of the decoding operation and achieved significant inference speedup. Galvez et al. (2024) optimized the decoding algorithm from a hardware perspective and minimized the GPU idle time during decoding, further improving GPU-based inference. Multi-blank Transducers (Xu et al., 2022) and Token-and-Duration Transducers (TDT) (Xu et al., 2023b) identified large numbers of blank predictions during RNN-T inference, proposing duration-prediction mechanisms to skip frames during processing. This improvement greatly reduced the number of decoding steps required for inference. While these methods improved Transducer inference speeds, the resulting models remained autoregressive. As such, Transducer inference speeds still significantly lagged behind non-autoregressive frameworks (e.g. CTC).

Our proposed semi-autoregressive method falls under the category of "fast/slow" methods, where the model combines a faster but less accurate module, and a slower but more accurate module to achieve good accuracy/efficiency tradeoffs. Inaguma et al. (2021) use a fast CTC model to generate multiple hypotheses, to be re-ranked by a slower AR model to improve speech translation performance. Mahadeokar et al. (2022) propose a cascade model architecture, where the fast model uses a relatively shallow encoder for quicker results, while the slow model uses additional encoder layers along with a separate joint network to improve results. Arora et al. (2024) proposed a model that divides speech input into blocks, and adopts AR processing among blocks but NAR processing within a block to balance speed with accuracy. Deliberation methods Hu et al. (2020) and its variants Wang et al. (2022) use cascaded encoders and separate decoder, and use the shallow encoder to generate

initial hypotheses, which to be fed into a second-pass "Listen-Attend-Spell" (LAS) model with additional encoder and decoders to improve accuracy. Compared with those "fast-slow" works, our work is unique in that our "slow" model only adds marginal parameters on top of the "fast" model, and the "refinement" process requires minimal computation overheads.

# 3 BACKGROUND

In this Section, we provide the technical background for models that influenced HAINAN architecture: CTC, RNN-Transducer (RNN-T), and Token-and-Duration Transducer (TDT).

## 3.1 CONNECTIONIST TEMPORAL CLASSIFICATION (CTC)

CTC (Graves et al., 2006) is a sequence-to-sequence prediction model designed to address the challenge of aligning input and output sequences of different lengths. Let $x = (x_1, x_2, ..., x_T)$ represent the input sequence, $y = (y_1, y_2, ..., y_U)$ the output sequence, and $\mathcal{V}$ the vocabulary, where $v \in \mathcal{V}$ includes all tokens in $y$ and a special "blank" symbol. A CTC model predicts $P(v|x_t)$ for each time step $t$. The "blank" symbol is crucial for handling the discrepancy between input and output sequence lengths, allowing for many-to-one alignments and representing frames that do not contribute additional information to the output. CTC models employ a set of rules to generate *CTC-augmented sequences* that are equivalent to the original sequence: (1) When a token is repeated in the original sequence, at least one "blank" token must be inserted between the repeated tokens, and (2) Any token can be repeated an arbitrary number of times.

The training objective of CTC models is to maximize the probability of $y$ given the input $x$. This probability is defined as the sum of probabilities for all possible augmented sequences of the reference sentence that match the input sequence length. CTC adopts a *conditional independence assumption* when computing the probability of an augmented sequence given the input. This assumption posits that each token in the augmented sequence is conditioned solely on the encoder output frame at its corresponding time step. Consequently, this allows for fully parallelized computation of $P(v|t)$ across different time steps, resulting in highly efficient inference.

CTC models with non-autoregressive components like self-attention and convolution are widely accepted as non-autoregressive models (Higuchi et al., 2021; 2020; Xu et al., 2023a), although during inference, CTC's token deduplication and blank handling would still depend on the last emitted token. We point out that this NAR categorization is based on the neural network computation of the model. In CTC, all neural network computation during inference can be carried out non-autoregressively, and the only autoregression is done on basic data types (strings, lists, etc.), which induces negligible computation compared to the costly neural network computations. In this work, we also follow this principle in categorizing models as autoregressive or not.

## 3.2 RECURRENT NEURAL NETWORK TRANSDUCER (RNN-T)

The RNN-Transducer, proposed by Graves (2012), represents a significant advancement in sequence-to-sequence modeling. RNN-T comprises three key components: an encoder, a predictor, and a joint network. The encoder extracts high-level features from the audio input, while the predictor processes the text history. The joint network then combines these outputs to generate a probability distribution over the vocabulary.

Like CTC, RNN-T incorporates a blank symbol, but with different rules for augmented sequences. (1) RNN-T sequences add zero or more blanks between adjacent tokens in the original sequence, rather than permitting repetition of both blanks and tokens, and (2) Only blank tokens consume input frames; the time step $t$ does not increment with non-blank token predictions of the model.

RNN-T's training aligns with CTC, maximizing the probability of the target sequence by summing over the probabilities of all possible augmented sequences. However, RNN-T diverges from CTC's conditional independence assumption. Instead, it leverages the predictor's output, representing text history, in the joint network to compute output distributions. As such, its predictions are conditioned by the alignment of input time step and position of current text output, making token estimation $P(v|x_t, y_{<u-1})$. This incorporation of text history enables RNN-T models to outperform CTC.

Despite its superior performance, RNN-T's reliance on the predictor and text history introduces autoregressive behavior in inference. Consequently, RNN-T inference resists full parallelization, resulting in lower efficiency compared to CTC. This trade-off between performance and efficiency highlights the ongoing challenge of designing optimal sequence-to-sequence models for ASR.

### 3.3 TOKEN-AND-DURATION TRANSDUCERS (TDT)

Token-and-Duration Transducer (TDT) model is introduced by Xu et al. (2023b). Unlike traditional Transducers where only blank predictions increment the encoder index $t$ by exactly one, TDT supports advancing $t$ by multiple steps for both blank and non-blank predictions. A TDT model generates two probability distributions, one for token, and the other for durations, which determines the number of audio frames to skip. By supporting frame skipping, TDT models require significantly fewer decoding steps compared to traditional Transducers, resulting in faster inference.

## 4 HYBRID-AUTOREGRESSIVE INFERENCE TRANSDUCER (HAINAN)

We propose *Hybrid-Autoregressive Inference Transducers* (HAINAN), an extension of the TDT model designed to support multiple modes of autoregression in inference. HAINAN incorporates *stochastic predictor masking*: during training, the predictor's output is randomly masked before being passed to the joint network. We implement this with a masking probability of 0.5, sampled at the `[batch, text-index]` level. Except for the random masking performed in training, the rest of the model training of HAINAN (computation in other components, and loss computation, etc) remains identical to that of TDT training. As a result of this change, during training, half the time the joint network is learning to predict with the whole model, and the other half the time it is learning to predict with information from the encoder only. Therefore when the model is well-trained, the joint network can generate valid distributions regardless of whether the predictor output is provided.

The HAINAN model supports three inference modes: autoregressive (AR), non-autoregressive (NAR) depending on whether we use the predictor network during inference. In addition, we also propose semi-autoregressive (SAR) inference, which combines elements of both NAR and AR inference to achieve better speed-accuracy tradeoff, and a Viterbi-based inference, which leverages the better representations learned by HAINAN models for better accuracy.

---

**Algorithm 1** Autoregressive Inference

1: **Input:** acoustic input $x$
2: enc = encoder($x$) # [T, H]
3: hyp = []; $t = 0$
4: **while** $t <$ len(enc) **do**
5:     pred = predictor(hyp) # [H]
6:     token_probs, dur_probs = joint(enc[$t$], pred) # [V] and [D]
7:     token = argmax(token_probs)
8:     duration = argmax(dur_probs)
9:     **if** token is not blank **then**
10:         hyp.append(token)
11:     $t$ += duration
12: **Return** hyp

**Algorithm 2** Non-AR Inference

1: **Input:** acoustic input $x$
2: enc = encoder($x$) # [T, H]
3: token_probs, dur_probs = parallel_joint(enc, pred=None) # [T, V] and [T, D]
4: tokens = argmax(token_probs, dim=-1) # [T]
5: durations = argmax(dur_probs, dim=-1) # [T]
6: hyp = []; $t = 0$
7: **while** $t <$ len(enc) **do**
8:     token = tokens[t]
9:     **if** token is not blank **then**
10:         hyp.append(token)
11:     $t$ += max(1, duration[t]) # avoid infinite loop
12: **return** hyp

---

### 4.1 AUTOREGRESSIVE INFERENCE

Autoregressive inference of HAINAN models is identical to TDT. For ease of reference, we provide Algorithm 1, which is a rewrite of the original Algorithm 2 from Xu et al. (2023b) with slightly changed notation/naming conventions to be consistent with this paper. To facilitate understanding of the Algorithms, we include tensor shapes as comments in the Algorithm, and we use T, V, D, U, H to represent audio-length, num-tokens, num-durations, text-index, and hidden-dimensions.

## 4.2 Non-autoregressive inference

Algorithm 2 shows NAR inference procedure. Compared to AR inference, NAR inference requires simple modifications of (1) passing an all-zero tensor instead of predictor output to the joint network; (2) moving the joint network computation out of the decoding loop, allowing parallel processing of all encoder outputs; (3) after each step, we increment $t$ by at least one to avoid infinite loops.

## 4.3 Semi-autoregressive inference

SAR inference is a novel algorithm combining elements of both AR and NAR approaches, as detailed in Algorithm 3. For ease of description, we assume the NAR inference procedure also returns the corresponding time-stamps of token outputs. The Algorithm consists of two main phases:

1. **Initial Hypothesis Generation (lines 2-4)** performs NAR inference to generate an initial hypothesis. It also uses the time stamps of the initial hypothesis to extract token-emitting frames from the encoder output for later processing.

2. **Hypothesis Refinement (lines 5-8)** uses the predictor to compute representations of all partial history contexts in the current hypothesis. It then combines the predictor's output with selected encoder output frames in the joint network to re-compute probability distributions for those frames, in parallel[2]. This procedure can be repeated to further improve the results. Note if repeated, all but the last round of the refinement should limit the argmax on line 9 to non-blank tokens since the token would need to be passed in the predictor for the next round of refinement.

The SAR approach combines the strengths of AR and NAR methods, allowing high degrees of parallelization while leveraging text histories in predictor computation to improve model accuracy.

---

**Algorithm 3** Semi-Autoregressive Inference of HAINAN Models

---

1: **Input:** acoustic input $x$
2: enc = encoder(x) # [T, H]
3: hyp, time_stamps = NARInference(enc) # [U] and [U]
4: useful_frames = enc[time_stamps,:] # [U, H]
5: shifted_hyp = [BOS] + hyp[:-1] # [U]
6: pred = predictor(shifted_hyp) # [U, H]
7: token_probs, duration_probs = parallel_joint(useful_frames, pred) # of shapes: [U, V] and [U, D]
8: hyp = argmax(token_probs, dim=-1) # [U]
9: **Return:** hyp

---

## 4.4 Viterbi-decoding

In addition to AR/NAR and SAR inference, we also propose a Viterbi-based inference method for HAINAN models. Given an audio sequence of length $T$, we view the output of the NAR HAINAN model as a *directed acyclic graph* (DAG) with $T + 1$ nodes, while the first $T$ correspond to frames and the last acts as a special end-of-sentence (eos) token. For each frame, we only consider the argmax token probability, which represents the weight of the first $T$ nodes (the eos node has a weight 1). The duration outputs at each frame correspond to transition weights into neighboring nodes. For example, in our models with max-duration 8, each node has at most 8 incoming and 8 outgoing connections connecting to its neighbors.

With this setting, speech recognition on the input audio is equivalent to finding the best path of the DAG from the first node to the last, which can be efficiently solved by a simplified Viterbi decoding algorithm. Notably, Viterbi and SAR techniques are orthogonal, and after Viterbi decoding generates the tokens and time stamps, we can use SAR techniques to further improve the results. The Algorithm for Viterbi decoding is shown in Algorithm 4. For simplicity, this Algorithm returns a "backtrack" structure that encodes the selected frames found by the best-path algorithm. The Algorithm's key is between lines 7 and 14, where nested loops run over (time, duration) pairs, and for each node, it finds the probability of the best path from the first node to that node using a dynamic

---

[2]The computation at line 7 is done fully parallelized, and line 6 may require some sequential computation if the predictor is an LSTM. If the predictor is stateless, this computation can also be fully parallelized as well.

programming algorithm. The Algorithm can be carried out efficiently with $O(TD)$ time complexity, where $T$ is the input sequence length and $D$ is the number of durations. In terms of space complexity, this Viterbi algorithm requires $O(T)$ (to store best-probs and backtrace pointers for each time index). Because duration 0 is not allowed in NAR modes, we assume there's no duration 0 in $N$.

---

**Algorithm 4** Viterbi Decoding of HAINAN Models

---

1: **Input:** acoustic input $x$, supported durations $N$
2: enc = encoder(x)
3: token_probs, duration_probs = parallel_joint(enc, pred=None) # of shapes [T, V] and [T, D]
4: best_prob_per_frame = max(token_probs, dim=-1) # of shape [T]
5: best_prob = [0.0 for t in range(len(x) + 1)]
6: backtrack = [-1 for t in range(len(x) + 1)] # we include a terminal node for backtracing
7: **for** target in range(1, len(x) + 1) **do**
8:    **for** idx, n in enumerate(N) **do**
9:       source = max(t-n, 0) # duration states cannot originate before initial node
10:       alpha = best-prob[source]
11:       trans_prob = best_prob_per_frame[target] * duration_prob[source,idx]
12:       **if** alpha * trans_prob > best_prob[target] **then**
13:          best_prob[target] = alpha * trans_prob
14:          backtrack[target] = source
15: **Return:** backtrack

---

## 5 EXPERIMENTS

We evaluate HAINAN ASR in two languages: English, German. All experiments are conducted using the NeMo (Kuchaiev et al., 2019) toolkit, version `1.23.0`. We first extract 80-dimensional filterbank features on audio, with 25 ms windows at 10 ms strides. All models use FastConformer encoders with the first three layers all performing 2X subsampling, therefore 8X subsampling in total. Both TDT and HAINAN models use durations $\{0, 1, 2, ..., 7, 8\}$. BPE tokenizer (Sennrich et al., 2016; Kudo & Richardson, 2018) of size 1024 is used for text representation. For all experiments, we let models train for sufficient steps until validation performance degrades (no more than 150k training steps), and run model averaging on 5 best checkpoints to generate the final model for evaluation. For all datasets, we report ASR performance measured by WER and the inference speed of RNN-T, CTC, TDT and HAINAN models. For HAINAN models, we include results of AR, NAR and SAR modes, and use the notation "SAR-$n$" to indicate $n$ rounds of refinement runs). We include Transducer models with stateless predictors with 1-token context denoted as "stateless", and use "/0" to denote TDT/HAINAN models that use all positive durations $\{1, 2, ..., 7, 8\}$.

### 5.1 ENGLISH ASR

We train our English models on the combination of Librispeech (Panayotov et al., 2015), Mozilla Common Voice (Ardila et al., 2019), VoxPopuli (Wang et al., 2021), Fisher (Cieri et al., 2004), People's Speech (Galvez et al., 2021), Wall Street Journal (Paul & Baker, 1992), National Speech Corpus (Koh et al., 2019), VCTK (Yamagishi et al., 2019), Multilingual Librispeech (Pratap et al., 2020), Europarl (Koehn, 2005) datasets, plus Suno AI datasets. In total, our dataset consists of approximately 60,000 hours of speech. The encoder uses FastConformer-XXL architecture with 42 layers of conformer blocks, each of which uses 8 heads of self-attention layers with model hidden dimension = 1024, totaling around 1.1b parameters. The convolutions in the conformers use kernel size = 9. The encoder was initialized with the public checkpoint `https://hf.co/nvidia/parakeet-tdt-1.1b`. For standard RNN-T, TDT and HAINAN models, their predictors consist of 2-layer LSTMs, with hidden dimension = 640. The joint network is a 2-layer feed-forward network with ReLU in between and with hidden dimension 1024.

We evaluate the models using the Huggingface ASR leaderboard (Srivastav et al., 2023), including the following datasets: AMI test (ami), Earnings22 test (e22), Gigaspeech test (giga), Librispeech test-clean and test-other (clean and other), Spgispeech test (spgi), Tedlium test (ted), and VoxPopuli test (vox). The results are shown in Table 1. Since there are multiple datasets, we direct readers' attention to the average WER achieved by different models. From the Table we can see:

Table 1: HAINAN English ASR: accuracy (WER%) and wall time on Huggingface ASR leaderboard datasets. AR/NAR/SAR represent autoregressive/non-autoregressive and semi-autoregressive. The number after SAR represents the number of refinement runs. Decoding time (seconds) is measured on only librispeech-test-other using batch=1 and beam=1, running on 2 A6000 GPUs. Viterbi time is not included since it's not comparable due to the different nature of the search algorithm. All models have around 1.1 billion parameters differing by at most 600k in actual number.

| model | AR | ami | e22 | giga | clean | other | spgi | ted | vox | **AVG** | time |
|---|---|---|---|---|---|---|---|---|---|---|---|
| RNN-T | AR | 16.90 | 13.87 | 9.76 | 1.43 | 2.75 | 3.40 | 3.63 | 5.49 | 7.15 | 179 |
| TDT | AR | 16.18 | 14.62 | 9.61 | 1.39 | 2.53 | 3.47 | 3.75 | 5.52 | 7.13 | 88 |
| TDT/0 | AR | 16.00 | 14.64 | 9.61 | 1.38 | 2.61 | 3.36 | 3.67 | 5.50 | 7.10 | 87 |
| stateless TDT | AR | 15.96 | 14.43 | 9.63 | 1.36 | 2.59 | 3.37 | 3.59 | 5.58 | 7.06 | 84 |
| CTC | NAR | 16.59 | 14.43 | 9.93 | 1.53 | 2.93 | 3.95 | 3.84 | 5.81 | 7.38 | 39 |
| HAINAN | AR | 15.92 | 14.16 | 9.70 | 1.46 | 2.76 | 3.59 | 3.59 | 5.60 | 7.10 | 89 |
| | NAR | 16.13 | 14.38 | 9.79 | 1.49 | 2.83 | 3.61 | 3.64 | 5.68 | 7.19 | 41 |
| | +Viterbi | 16.23 | 14.03 | 9.75 | 1.48 | 2.81 | 3.53 | 3.60 | 5.60 | 7.13 | - |
| | SAR-1 | 16.01 | 14.23 | 9.73 | 1.47 | 2.76 | 3.60 | 3.64 | 5.61 | 7.13 | 45 |
| | +Viterbi | 16.13 | 13.93 | 9.71 | 1.47 | 2.77 | 3.51 | 3.60 | 5.56 | 7.09 | - |
| | SAR-2 | 15.99 | 14.17 | 9.72 | 1.47 | 2.75 | 3.59 | 3.63 | 5.61 | 7.12 | 48 |
| stateless HAINAN | AR | 15.69 | 13.64 | 9.68 | 1.49 | 2.82 | 3.47 | 3.74 | 5.72 | 7.03 | 82 |
| | NAR | 15.78 | 13.73 | 9.77 | 1.49 | 2.91 | 3.49 | 3.75 | 5.75 | 7.08 | 40 |
| | +Viterbi | 15.87 | 13.45 | 9.72 | 1.47 | 2.87 | 3.42 | 3.74 | 5.70 | 7.03 | - |
| | SAR-1 | 15.77 | 13.72 | 9.72 | 1.49 | 2.84 | 3.48 | 3.74 | 5.71 | 7.06 | 42 |
| | +Viterbi | 15.86 | 13.42 | 9.69 | 1.47 | 2.82 | 3.41 | 3.74 | 5.69 | 7.01 | - |
| | SAR-2 | 15.75 | 13.61 | 9.70 | 1.48 | 2.84 | 3.47 | 3.72 | 5.71 | 7.03 | 43 |
| HAINAN/0 | AR | 15.75 | 13.37 | 9.73 | 1.40 | 2.83 | 3.52 | 3.56 | 5.76 | 6.98 | 92 |
| | NAR | 15.94 | 13.67 | 9.81 | 1.44 | 2.87 | 3.55 | 3.54 | 5.74 | 7.07 | 40 |
| | +Viterbi | 15.88 | 13.38 | 9.73 | 1.41 | 2.86 | 3.48 | 3.48 | 5.65 | 6.98 | - |
| | SAR-1 | 15.84 | 13.45 | 9.74 | 1.41 | 2.85 | 3.52 | 3.54 | 5.69 | 7.00 | 44 |
| | +Viterbi | 15.79 | 13.21 | 9.68 | 1.39 | 2.83 | 3.46 | 3.50 | 5.60 | 6.93 | - |
| | SAR-2 | 15.79 | 13.36 | 9.73 | 1.40 | 2.83 | 3.52 | 3.55 | 5.69 | 6.98 | 48 |

- Regardless of the nature of the predictor, AR HAINAN models achieve superior accuracy than RNN-T and TDT counterparts, with similar efficiency.

- NAR-HAINAN achieves consistently better accuracy than CTC, with similar inference speed. In particular, NAR stateless HAINAN achieves better accuracy than the AR TDT model, while running over 2X faster; it also outperforms RNN-T model, while being over 4X faster.

- SAR further improves the accuracy of NAR inference with small computational overhead. With standard HAINAN, each additional SAR refinement run adds around 3-4 seconds of processing for the whole dataset, while stateless HAINAN only requires 1-2 seconds per SAR run. This is significantly faster than fully AR TDT runs which takes around 40 seconds more processing time in total. Also in both models, just one round of SAR makes the model catch up with TDT performance while running at least 2X faster.

- Viterbi decoding can further improve model accuracy. We present an analysis in subsection 6.2 that can explain this phenomenon.

- In particular, the HAINAN model with non-zero durations achieve the best results overall. Later in Section 6.4, we present our discovery that can explain this phenomenon.

## 5.2 GERMAN ASR

Our German models are trained on German Mozilla Common Voice, Multilingual Librispeech, and VoxPopuli datasets, totaling around 2000 hours. We use FastConformerLarge configuration of around 110m parameters [3], and evaluate the models on the German Multilingual Librispeech (MLS) and VoxPopuli (Vox) testsets. For training, all model encoders are initialized with public

---

[3]The hyper-parameters can be found at `https://github.com/NVIDIA/NeMo/blob/main/examples/asr/conf/fastconformer/fast-conformer_transducer_bpe.yaml`

checkpoint `https://hf.co/nvidia/stt_en_fastconformer_transducer_large`. The results of those models are shown in Table 2.

We see similar trends compared to our English models, where HAINAN achieves similar AR accuracy and efficiency with TDT models, and significantly improved NAR accuracy compared to CTC models, with similar inference speed. SAR further closes the accuracy gaps between AR and NAR, while adding small overheads, and still being around 4X faster than AR inference.

Table 2: German ASR: accuracy (WER%) and inference wall time (seconds) on Multilingual Librispeech and VoxPopuli. Decoding time is measured with batch=1 and beam=1.

| model | AR | MLS | time | VOX | time |
|---|---|---|---|---|---|
| RNN-T | AR | 4.90 | 429 | 9.47 | 113 |
| TDT | AR | 4.77 | 141 | 9.41 | 52 |
| TDT/0 | AR | 4.86 | 140 | 9.47 | 52 |
| stateless TDT | AR | 4.88 | 131 | 9.36 | 49 |
| CTC | NAR | 5.92 | 22 | 10.25 | 8 |
| HAINAN | AR | 4.75 | 141 | 9.22 | 52 |
| | NAR | 5.11 | 20 | 9.63 | 8 |
| | SAR-1 | 4.87 | 29 | 9.47 | 11 |
| | SAR-2 | 4.83 | 36 | 9.44 | 14 |
| stateless HAINAN | AR | 4.85 | 127 | 9.35 | 46 |
| | NAR | 5.09 | 22 | 9.57 | 8 |
| | SAR-1 | 4.89 | 24 | 9.45 | 9 |
| | SAR-2 | 4.87 | 25 | 9.44 | 10 |
| HAINAN/0 | AR | 4.92 | 140 | 9.59 | 53 |
| | NAR | 5.37 | 22 | 10.08 | 8 |
| | SAR-1 | 5.12 | 29 | 9.87 | 11 |
| | SAR-2 | 5.05 | 35 | 9.84 | 14 |

# 6 ANALYSIS AND DISCUSSIONS

## 6.1 LIMITATIONS OF SAR METHODS

The key of SAR method is to reuse the time stamps generated in the initial NAR inference, and only regenerates the token outputs for those frames. This puts limitations on the types of errors that can be corrected by the method. We provide some analysis here.

- From the subword level, SAR can only fix substitution errors by replacing a wrong subword with the correct one, and insertion errors by replacing the inserted wrong subword with a blank. It can't fix deletions where the hypothesis is shorter than the reference.
- From the word level, SAR can correct deletion errors. Note, although SAR can't add more subwords to the original output, it can replace a non-word-beginning subword with a word-beginning subword, and thus increase the number of words in the SAR output. E.g. the SAR can change "forty" to "for tea", by replacing subwords "_for ty" with "_for _tea", where "_" means the begin-of-word symbol.

Table 3: Breakdown of Error types (%) of German HAINAN Models on Voxpopuli Dataset.

| inference | sub | del | ins | total |
|---|---|---|---|---|
| NAR | 5.19 | 2.64 | 1.80 | 9.63 |
| SAR-1 | 5.00 | 2.85 | 1.62 | 9.47 |
| SAR-2 | 4.96 | 2.87 | 1.61 | 9.44 |

We report detailed breakdown of error types in our German HAINAN model on Voxpopuli comparing NAR and SAR in Table 3. We see that SAR models primarily reduce substitution and insertion

errors; in fact, the total amount of deletion errors increased with SAR models. Even though the overall deletion error numbers go up, we do observe in certain utterances, SAR can correct deletion errors from NAR outputs. We provide some examples in Appendix subsection A.3.

## 6.2 NAR HAINAN'S CAPABILITY OF CAPTURING ACOUSTIC AMBIGUITY

In this subsection, we highlight an aspect of NAR HAINAN model that grants it greater modeling power compared CTC models: its ability to support more diverse model representation and multiple hypotheses across different time stamps, especially for potentially ambiguous speech. To illustrate this, we present a real-world example using an audio recording of the phrase "ice cream". First, we run our English CTC model on the utterance to extract the arg-max token per frame, resulting in:

```
     i   ce   cre   am    
```

The CTC model output correctly represents the "ice cream" hypothesis, with each non-blank token appearing once, surrounded by blanks. This aligns with the well-documented *peaky behavior* of CTC (Zeyer et al., 2021). In contrast, the arg-max tokens emitted from NAR HAINAN model are:

```
 i i i i i ce ce sc sc cre cre re am am am   
```

Notably, the per-frame arg-max output not only encodes "ice cream" but also captures an alternative hypothesis: "i scream". These hypotheses are dispersed across different time frames. For clarity, we have color-coded the tokens: orange for "ice cream", blue for "i scream", and green for shared tokens. It's worth noting that these hypotheses are acoustically similar, and their different alignments to the audio demonstrate that the NAR HAINAN's per-frame arg-max output provides a more accurate representation of the audio input. We believe this is the reason why the Viterbi algorithm can achieve improved accuracy.

## 6.3 IMPACT OF DURATION CONFIGURATIONS

In this subsection, we examine how duration configurations affect HAINAN model performance. We vary the max-durations in our German models and report their ASR performance in Table 4. Our results reveal a clear trend: models with max-durations of 1 and 2 perform significantly worse, while 4 and 8 yield improved results. This observation aligns with our earlier analysis, suggesting that shorter maximal durations constrain the model's ability to learn complex structures, and the max duration needs to be sufficiently large to represent the real distribution. Interestingly, max-duration 4 provides the best overall results, despite our choice of max-duration 8 for most experiments. We selected the latter to accommodate greater linguistic variability across datasets, as we did not extensively tune this parameter.

## 6.4 DURATION DISTRIBUTION ANALYSIS

Building on our previous analysis of maximum duration settings, we now examine the actual duration predictions made by our models. This analysis provides insights into how the models utilize the allowed duration range and how the HAINAN training procedure affects duration prediction behavior. We conducted experiments using HAINAN and TDT (trained with max-duration=8), running different types of inference on the Librispeech test-other and German Multilingual Librispeech test

Table 4: German ASR results of NAR HAINAN models with different maximum durations.

| max-duration | MLS | VOX |
|---|---|---|
| 1 | 7.58 | 12.46 |
| 2 | 5.79 | 10.27 |
| 4 | 5.06 | 9.56 |
| 8 | 5.09 | 9.57 |

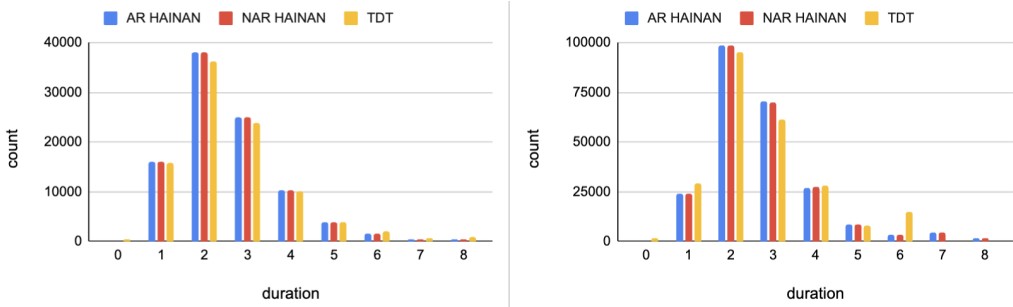

Figure 1: Duration prediction count of HAINAN and TDT models on Librispeech test-other (left) and German Multilingual Librispeech test (right).

sets. Figure 1 shows the distribution of duration predictions for HAINAN (in both AR and NAR modes) compared to a standard TDT model. The results reveal several interesting patterns:

- **Zero-duration mitigation:** HAINAN's training procedure dramatically reduces the occurrence of 0 duration predictions, effectively addressing the potential issue of infinite loops in NAR inference. This phenomenon explains the results from Table 1, where excluding duration 0 gives the best results among all models.

- **Preference for shorter durations:** Despite allowing durations up to 8, all models show a preference for durations of 4 or less, with the preference stronger with HAINAN models. This aligns with our previous finding that max-duration=4 tends to yield better performance.

- **Consistency across inference modes:** HAINAN shows remarkably similar duration distributions in AR and NAR modes, suggesting a robust learning of duration prediction strategies.

These findings demonstrate that while allowing longer maximum durations (e.g., 8) provides flexibility, the HAINAN model naturally learns to favor shorter durations. This behavior reconciles with our earlier observation that max-duration=4 yields optimal performance: the model has the flexibility to use longer durations when necessary but primarily operates within the 1-4 range.

## 7 CONCLUSION AND FUTURE WORK

In this paper, we introduced **H**ybrid-**A**utoregressive **IN**ference TrANsducers (HAINAN), a novel approach that aims to bridge the gap between autoregressive and non-autoregressive methods in speech recognition. By implementing a simple stochastic masking technique during training, we've developed a model that offers flexibility in its inference strategy. Our experiments across multiple languages demonstrate promising results:

- In AR mode, HAINAN achieves on-par performance with TDT and RNN-T models.
- In NAR mode, HAINAN achieves significantly improved performance to CTC models.
- We propose a semi-autoregressive (SAR) mode, a new approach to balancing speed and accuracy.
- We also propose a simplified Viterbi decoding method to further improve model accuracy.

There are several future research directions, including applying our method to other speech and language-related tasks including speech translation, spoken language understanding, and speech synthesis, etc. We plan to investigate more flexible model architectures further to bridge the gaps between AR and NAR models. Furthermore, we plan to further improve the semi-autoregressive algorithm for better speed-accuracy balancing.

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

# A APPENDIX

## A.1 TRAINING ALIGNMENTS

The strength of NAR HAINAN to better model acoustic ambiguity can also be observed during model training. Figure 2 shows the force-alignment probabilities for both "ice cream" and "i scream" hypotheses. Notably, the HAINAN model can align both hypotheses with high probabilities by separating them at different time stamps, a capability not possible with CTC models due to their frame-by-frame independence assumption and lack of frame-skipping mechanism.

To understand why CTC models cannot assign high probabilities to both hypotheses simultaneously, consider the following scenario: If one frame assigns high probability to "ce" (from "ice"), and another frame assigns high probability to "sc" (from "scream"), the resulting sequence would inevitably be recognized as "cesc" or "scce", matching neither "ice cream" nor "i scream". This limitation is unique to frame-by-frame models with conditional independence assumptions like CTC. In contrast, the HAINAN model's ability to skip frames and represent multiple hypotheses allows it to better capture the inherent ambiguity in speech.

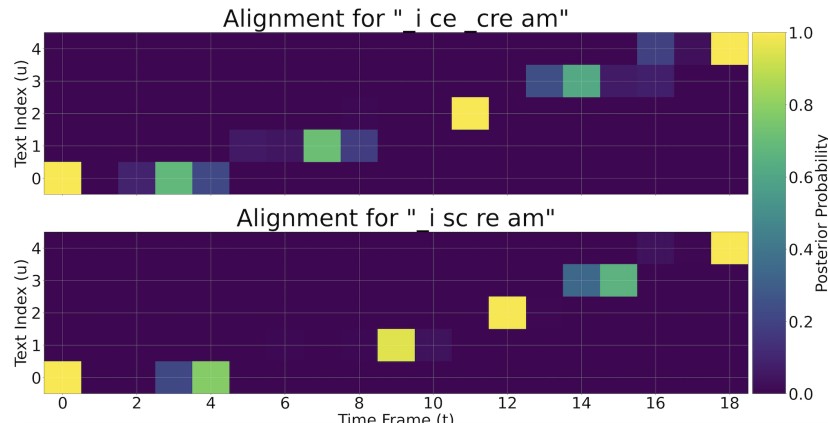

Figure 2: Alignments generated by HAINAN model without predictor, on "ice cream" and "i scream".

## A.2 COMPARISON WITH HYBRID-TDT-CTC MODELS

To further validate the effectiveness of our approach, we compare HAINAN with Hybrid-TDT-CTC models (Koluguri et al., 2024), which combines elements of both CTC and TDT models. With the Hybrid-TDT-CTC model, the encoder output is processed by both a CTC embedding layer and TDT joint/predictors, and the model is trained using a linear interpolation of CTC and TDT losses. During inference, the model supports both AR (using TDT components) and NAR (using CTC components) inference modes. Table 5 presents the performance comparison between Hybrid-TDT-CTC and HAINAN models across various datasets from the Huggingface ASR leaderboard.

Key observations from this comparison:

Table 5: Hybrid-TDT-CTC model accuracy (WER%) on Huggingface ASR leaderboard datasets

| model | inference | ami | e22 | giga | clean | other | spgi | ted | vox | **AVG** |
|---|---|---|---|---|---|---|---|---|---|---|
| hybrid | TDT (AR) | 16.17 | 14.77 | 9.56 | 1.37 | 2.64 | 3.65 | 3.71 | 5.56 | 7.18 |
| | CTC (NAR) | 17.13 | 15.33 | 9.90 | 1.51 | 2.88 | 3.65 | 3.95 | 5.71 | 7.51 |
| HAINAN | AR | 15.92 | 14.16 | 9.70 | 1.46 | 2.76 | 3.59 | 3.59 | 5.60 | 7.10 |
| | NAR | 16.13 | 14.38 | 9.79 | 1.49 | 2.83 | 3.61 | 3.64 | 5.68 | 7.19 |

- **Performance:** HAINAN outperforms Hybrid-TDT-CTC in both AR and NAR modes across most datasets, with lower average WER scores.

- **Model Efficiency:** Unlike Hybrid-TDT-CTC, HAINAN does not require storing separate sets of parameters for AR and NAR inference, leading to a more compact model.

- **Flexibility:** HAINAN offers an additional semi-autoregressive (SAR) inference mode, providing a more flexible speed-accuracy trade-off not available in Hybrid-TDT-CTC models.

These results underscore the advantages of HAINAN's unified architecture, which achieves superior performance while maintaining model simplicity and offering greater inference flexibility.

### A.3 EXAMPLES OF DELETION ERRORS CORRECTED BY SAR METHODS

Although relatively rare, SAR methods can correct deletion errors of NAR inference results. Here are some examples in German and English that we observe in our runs. We use green to highlight words that were missing in NAR hypotheses but included by SAR.

German:

Table 6: Example of German ASR where SAR reduces deletion errors from NAR output

| | |
|---|---|
| reference | die meisten vorfälle passieren bei der implementierung bedauerlicherweise im krankenhaus häufig auch in der anwendung |
| NAR | diesfälle passieren bei der implementierung bedauerlicherweise im krankenhaus häufig auch in der anwendung |
| SAR | die handfälle passieren bei der implementierung bedauerlicherweise im krankenhaus häufig auch in der anwendung |
| reference | ich ersuche den kommissar hier sofortmaßnahmen zu ergreifen |
| NAR | ichsuche den kommissar hier sofort maßnahmen zu ergreifen |
| SAR | ich versuche den kommissar hier sofortmaßnahmen zu ergreifen |

English:

Table 7: Example of English ASR where SAR reduces deletion errors from NAR output

| | |
|---|---|
| reference | then one of them says kind of soft and gentle |
| NAR | then one of them says kinda soft and gentle |
| SAR | then one of them says kind of soft and gentle |
| reference | say mester gurr sir which thankful i am to you for speaking so but you don't really think as he has come to harm |
| NAR | say mister gurrsir which thankful i am for you for speaking so but you don't really think as he has come to harm |
| SAR | say mister gurr sir which thankful i am for you for speaking so but you don't really think as he has come to harm |

We notice that, in all those examples of corrected deletion errors, the SAR method replaces a non-word-beginning subword with a word-beginning subword, e.g. "␣kind a" is replaced with "␣kind ␣of". This aligns with our analysis in subsection 6.1, since this is the only way to increase the number of words while keeping the number of subwords fixed.

### A.4 REASON OF ZERO-DURATION MITIGATION WITH HAINAN

In subsection 6.4, we show empirically that duration 0 is greatly mitigated by HAINAN model. In this subsection, we provide some analysis of why this is the case. In our description, we use $E_t$ and $D_u$ to denote encoder/predictor outputs.

1. By performing stochastic masking of predictor output during training, the model is learning to make the output distribution of $\text{joint}(E_t, D_u)$ similar to $\text{joint}(E_t, D_u * 0)$.

2. If duration 0 is predicted from $\text{joint}(E_t, D_u)$, then the next step of processing would be $\text{joint}(E_t, D_{u+1})$, where $t$ remains the same but $u$ gets incremented by 1.

3. From 1, we know that model is learning to make $\text{joint}(E_t, D_u)$ similar to $\text{joint}(E_t, D_u * 0)$; also it's learning to make $\text{joint}(E_t, D_{u+1})$ similar to $\text{joint}(E_t, D_{u+1} * 0)$. Consequently, the model would make $\text{joint}(E_t, D_u)$ similar to $\text{joint}(E_t, D_{u+1})$, since $\text{joint}(E_t, D_u * 0)$ is the same as $\text{joint}(E_t, D_{u+1} * 0)$, both of which has predictor masked out.

4. However, we know that $\text{joint}(E_t, D_u)$ and $\text{joint}(E_t, D_{u+1})$ should predict different tokens since the text histories are different. This causes a contradiction.

Therefore, HAINAN models, by adopting stochastic predictor masking, would naturally suppress the prediction of duration 0 in its processing.

