# OpenReview forum: "HAINAN: Fast and Accurate Transducer for Hybrid-Autoregressive ASR"
_ICLR.cc/2025/Conference — ICLR 2025 Poster_

### Official Review · Reviewer_pghP · 2024-10-28

**Soundness:** 2
**Presentation:** 3
**Contribution:** 2
**Rating:** 5
**Confidence:** 4

**Summary:**

This paper introduces a model called HAI-T, an extension of the Token-and-Duration Transducer (TDT). TDT itself is an extension of RNN-T, where token prediction and duration prediction are decoupled. The authors propose randomly dropping the predictor network of TDT so that the model can operate in both auto-regressive (AR) and non-autoregressive (NAR) modes. They also investigate a semi-AR mode, where the NAR decoding result is refined through AR decoding.

**Strengths:**

The paper presents a novel idea of randomly dropping the predictor network to enable the TDT model to function in both AR and NAR modes. Especially, as far as I am aware, this is the first work to evaluate the NAR version of the TDT model.

**Weaknesses:**

- Insufficient Contributions
  - The idea of randomly dropping the decoder network is quite simple, making this more of an investigation paper rather than a presentation of a highly novel idea. In this context, rigorous experimental validation is crucial, but the evaluation provided is not sufficient to substantiate their claims (see the next section).
  - The combination of NAR and AR modes -- or more generally, the combination of a faster, less accurate model with a slower, more accurate model -- has been explored extensively. The method proposed in this paper is relatively straightforward and lacks theoretical novelty. I have listed a few papers as examples, but there are many more relevant works. I suggest explicitly discussing these prior works and highlighting specific differences in methodology or performance.
    - NAR + AR:
      - H. Inaguma, et al., “Non-autoregressive end-to-end speech translation with parallel autoregressive rescoring,” 2021.
      - S. Arora, et al., “Semi-Autoregressive Streaming ASR with Label Context,” ICASSP 2024.
    - Fast + Slow:
      - J. Mahadeokar, “Streaming parallel transducer beam search with fast-slow cascaded encoders,” Interspeech, 2022.

- Insufficient Evaluation
  - The authors claim that the proposed method outperforms RNN-T and TDT in AR mode, but their experimental evidence is not convincing. In the first 7 rows of Table 1, the AR mode results for RNN-T, TDT, and HAI-T (the proposed model) average 7.15%, 7.13%, and 7.10%, respectively. The difference from 7.13% to 7.10% is too minor to definitively demonstrate HAI-T's superiority. Moreover, when examining individual test sets like “ami” and “e22,” the proposed model often underperforms compared to RNN-T and TDT. It is likely that there is no statistically significant difference between TDT and HAI-T in AR mode, and the observed difference may be due to random fluctuation (e.g., the trend could easily reverse with additional testing data). More extensive experiments with statistical significance tests over multiple evaluation sets are suggested.
  -  Additionally, while the authors show that the HAI-T model performs better with a “stateless” decoder or when excluding 0-duration configurations in the remaining 8 results of Table 1, the same configurations should have been applied to the baseline models for a fair comparison.
  - The authors claim that their proposed method outperforms CTC in NAR mode, but this is not convincingly demonstrated. In the NAR setting, the HAI-T model slightly outperforms CTC (7.38% vs. 7.19%) but with a slight increase in computational overhead (41 vs. 39 in time). While the increase in time is minor, the improvement in WER is also small, raising concerns that the improvement may simply be due to increased parameters.
  - Additionally, the paper does not define the term “time,” which presents another issue. Please explicitly define how "time" is measured (e.g., wall clock time, CPU time, number of operations) and under what conditions (e.g., hardware specifications, batch size) instead of merely mentioning Huggingface ASR leaderboard.
  - It would be beneficial to include the “time” metric in Table 3.

- Description Issues
  - The descriptions of their “code (page 4)” and “algorithm (page 5)” are inadequately defined.
    - The code on page 4 seems unnecessary. The algorithm is simple enough that additional explanation is not required.
    - The algorithm on page 5 relies on hidden assumptions known only to the authors. For example, (1) it is unclear what the output of the “joint” function is, and why the second output is omitted, and (2) the application of “dim=-1” in argmax assumes a specific shape of “token-probs,” which remains undefined. Please make the algorithm self-contained by appropriately defining each notion.
  - The decoding strategy used for NAR-mode is unclear.
    - From the explanation in Section 4.1, it appears that the authors still applied a decoding algorithm used for TDT even when the model is used in NAR mode. I have several questions:
      - I assume that the decoding algorithm is still left-to-right. If this is the case, can it still be called "NAR"? Please explicitly define the notion of NAR in the paper.
      - Is there any notion of a 'beam' in the decoding algorithm for HAI-T? This is a clarification question, and I guess the answer is no. However, if beam search decoding was used, the configuration needs to be specified. In addition, if all experiments were conducted without beam search, it is still highly recommended to evaluate each method using the beam search configuration, as it is the most widely used.
      - What decoding algorithm is used for CTC (best path, prefix search)? Can the author provide the impact of decoding algorithms (not only for CTC, but also for RNN-T with beam size) to understand the impact of it?
    - In Section 6.2, the authors suddenly introduce Viterbi-based decoding in the experimental section. They should have described the details of the decoding strategies (both the one in the main experiment and Viterbi decoding) in the proposed method section to clearly show the differences between them. It would have been better to explicitly describe Algorithm 2 of Xu et al. (2023) instead of merely referencing it, as the algorithm is not widely known within the community. Including the memory footprint is also highly recommended, as Viterbi decoding may require significantly more memory during inference, especially for long input durations.
  - The discussion in Section 6.3 is unclear to me. The authors claimed that "the ability to skip more frames enables the model to learn better representations" by presenting Table 4 with different max-duration settings, where the shorter max-duration setting provided worse results. However, isn't it just showing that the shorter max-duration is not able to capture the real duration distribution? I think that the only conclusion we could draw from Table 4 is "the max duration needs to be sufficiently large to represent the real distribution," and their claim that "the ability to skip more frames enables the model to learn better representations" cannot be drawn.

**Questions:**

Please address the concerns and questions raised in the weaknesses section.

---

> ### Author Response · Authors · 2024-11-23
> **Author response, Part 1**
>
> - Contribution and relation to prior works
>
> Thanks for bringing our attention to those papers that explored combining a fast and a more accurate model for better modeling performance, and we’d be happy to include discussions in our paper regarding the difference between our and their work.
>
> 1. Reranking vs refining (Comparing with H. Inaguma et all)
>
> A key difference is that those works are of the framework of using the faster model to generate multiple hypotheses, which are reranked by the more accurate model. Our work, however, uses the AR model to “refine” the NAR hypothesis directly.
>
> This difference has a significant impact on the speed of models. For the “refinement” technique, the SAR rescoring steps add around 5% - 10% computation time overhead on top of the NAR models; however, with the “reranking” technique, since it involves generating multiple hypotheses, the approach adds significant computation overheads, e.g. in H. Inaguma et al., it is at least 2X slower than NAR CTC models according to Tables I and II.
>
> Theoretically, our method is orthogonal to the reranking approach. In the future, we plan to investigate combining our SAR refinement and also the k-best hypotheses reranking to better improve models.
>
> 2. Other "semi-autoregressive" methods (S. Arora, et al.)
>
> Although with similar names, our work is very different from S. Arora, et al. Their work proposes a structure where speech is processed in blocks, and within each block, they use NAR processing, but between different blocks, they utilize AR processing.
>
> 3. Fast-slow method in other network components (cascaded encoders by J. Mahadeokar et al.)
>
> The work by J. Mahadeokar et al. is an application of fast inference and slow rescore methods done at the encoder level. They use a cascaded encoder design in the model, each with separate joint networks for prediction, and the whole model is jointly trained, balancing the loss computed from the outputs of two joint networks.
>
> In their case, the slower model has extra parameters in its encoder and a separate joint network. Therefore, the slower model still adds considerable extra computation to the processing on top of the fast model.
>
> In comparison, in our case, aside from the decoder network (with very small #params) that is unique to the slow model, all other parameters are shared among the fast and slow models, offering greater flexibility for our deployment model. Furthermore, with our SAR methods, our “slower” model only adds less than 10% computation time compared to the “fast” model, due to the parallelized nature of the added computation.
>
>
> Our primary contribution is that we propose a single model that operates in both NAR and AR modes, and each mode achieves better or on-par accuracy with alternative methods while maintaining efficiency. This means adopters of CTC/RNNT/TDT models can switch to using our method safely without risk.
>
> Additionally, our SAR method, in particular, is a fast-slow method that gets to invent new hypotheses directly, rather than the commonly used reranking method where the fast model rescores a set of hypotheses generated by the slower model. This has the benefit of adding minimal computational cost to the pipeline while consistently improving accuracy. We also hope this paradigm can inspire new works in the research field, allowing better speed/accuracy tradeoffs in speech/language-related models. Also, this method is orthogonal to existing reranking methods, so it’s possible to combine them to further improve results.

---

> > ### Author Response · Authors · 2024-11-23
> > **Part 2**
> >
> > - Regarding our evaluation
> >
> > We take the suggestion and will change our paper so it says our method achieves “on-par” performance in AR modes compared to alternatives.
> >
> > - Regarding using a stateless prediction network/positive durations for baseline.
> >
> > Sure, we would be happy to include those results in the revised paper. Note, that we have observed that replacing LSTM predictors with stateless networks (with short context like 1 token, as used in our HAI-T models), and enforcing it to increment by at least 1 for Transducers would cause performance degradations. This is also consistent with findings from the existing literature, e.g. (Rnn-Transducer with Stateless Prediction Network, Godsi et al., and FAST AND PARALLEL DECODING FOR TRANSDUCER by Kang et al. In those works, using stateless decoder/monotonic-constrained attention is seen as more of a speed-up at the cost of slightly degraded performance.
> >
> > - Regarding NAR/CTC speed difference.
> >
> > We thank the reviewer for noticing this slight difference in inference speed. Due to the slightly different implementations of CTC-related layers and Transducer-related layer implementation, it’s hard to make sure they’re the exact same size. E.g. in our English models, the #params for CTC and NAR-HAI-T model are different by around 600k, while both models have over 1.1 billion parameters, making the relative difference around 0.005%. We believe that it’s not likely that this small size difference causes the speed/accuracy difference. Our investigation indicates that this has to do with the implementation pipelines for CTC VS Transducer model types, since the two models inherit different parent classes and they have slightly different inference pipeline implementations. We’re working on revising the pipeline since, in theory, the two models should have very similar inference speeds.
> >
> > In the paper when measuring the time of inference, we mean wall time. We will revise our writing to make that clear, as well as include more info on the machine/GPU specs etc.
> >
> > - On the description of the code and algorithm
> >
> > We understand that, given the textual description of masking out the prediction network, the code diff block seems unnecessary. We will remove it.
> >
> > The joint() function output is two tensors “token_probs” and “duration_probs”. In the SAR algorithm, we only use the token_probs to refine the output token, so the duration_prob is not used during the processing; and the argmax(..., dim=-1) is performed on the vocabulary dimension of the output of joint, i.e. finding the most probable token at different locations. We will revise our paper to include extra information on the input/output of the functions referenced in the Algorithm block to provide better clarity.
> >
> > - Regarding the terminology of NAR:
> >
> > Our decoding process is indeed done from left-to-right, and there are some sequential dependencies through duration-based frame selection, so strictly speaking, the decoding procedure is not NAR.
> >
> > We would be happy to revise our writing and emphasize that when we call the model NAR, it specifically regards the model’s neural network computation. Note, that the AR/NAR categorization for models usually refers to the neural network computations but not the complete procedure. As a point of comparison, even CTC models - widely accepted as non-autoregressive (e.g many papers from https://scholar.google.com/scholar?q=CTC+non+autoregressive), have implicit sequential dependencies in their decoding (e.g., blank token handling and repeated token resolution only depend on the last emitted token), yet are considered NAR due to their parallel neural computations.
> >
> > - Nature of the search algorithm
> >
> > All experiments reported in the paper (CTC, TDT, and NAR-HAI-T) use a one-best greedy search with batch=1, without any external language models or any prefix constraints to guide the search. This choice is based on the consideration that, It is generally reported in the literature that beam-search on Transducer-based models only brings marginal improvements while adding several times of runtime (e.g. LABEL-LOOPING: HIGHLY EFFICIENT DECODING FOR TRANSDUCERS by Bataev et al.), therefore this work focuses on more cost-effective methods for improving model performance rather than relying on beam-search.
> >
> > That being said, we plan to work on a novel beam-search method for HAI-T models -- a “2D beam-search method” that investigates both beams in token predictions from each frame, as well as beams in duration predictions. This aligns with the observation in Sec. 6.1 and 6.2 where richer representations learned by HAI-T models can bring further gains to model performance.

---

> > > ### Author Response · Authors · 2024-11-23
> > > **Part 3**
> > >
> > > - original TDT decoding and Viterbi decoding algorithm
> > >
> > > Thanks for pointing out that referencing Algorithm 2 of Xu et al. might not be as good as explicitly describing it in the paper. We will include the algorithm details if space permits.
> > >
> > > As for the Viterbi algorithm, we purposefully put it in the analysis section rather than the main results. The aim of including those results is to show that the HAI-T models can learn more diverse representations among different time indices. Using Viterbi decoding is intended as an empirical validation of this claim, which we see from consistent accuracy improvements. It’s not an apple-to-apple comparison if include Viterbi results with our main table, since the nature of the search algorithms is different (we can think of the Viterbi algorithm as a special case of beam-search where the beam is limited in the duration outputs but not tokens).
> > >
> > > That being said, since multiple reviewers have suggested moving Viterbi to the main method section, we will re-organize the paper and make this change. If there’s space left, we will also include more analysis of the algorithm, e.g. the time and space complexity, as well as memory footprint.
> > >
> > > - The discussion in Section 6.3
> > >
> > > Thanks for pointing out the possible confusion in Section 6.3. We will revise it to make it more clear.
> > >
> > > To explain what we mean, allow us to first clarify what we mean by “better representation”. If we look at the two cases shown in Section 6.1, where the CTC’s per-frame output can only correspond to a single hypothesis, and the NAR HAI-T’s per-frame output corresponds to two alternative hypotheses, dispersed at different time stamps. The second one is considered a “better representation”, since the two alternative hypotheses successfully capture the acoustic similarities.
> > >
> > > While the conclusion that the reviewer suggested, “the max duration needs to be sufficiently large to represent the real distribution” is also correct, and is a direct conclusion from Table 4, we believe there are deeper reasons for that. The reason we claim “the ability to skip more frames” is because, e.g. with the example of “i i i i i ce ce sc sc cre cre re am am am”, such representation is only possible because the model can use the duration output to skip frames, so that the tokens in the two hypotheses won’t show up together in the model output. In this context, a CTC model is one that can’t skip frames, and therefore its per-frame output is less diverse, only representing one hypothesis.
> > >
> > > We plan to include both what the reviewer suggested as a direct conclusion for Table 4; and also combine the information from Table 4 and Section 6.1 to talk about the aspect of diverse representation brought by the ability to skip frames. Furthermore, we agree the reviewer’s wording of “sufficiently large” is a better way to describe our point: “NAR-HAI-T model’s ability to skip a sufficiently large number of frames …”. Since it’s not always the case that the more frames can be skipped, the better accuracy the model can achieve. We thank the reviewer again for raising this point and will update our paper accordingly.

---

> ### Comment · Reviewer_pghP · 2024-11-23
> **Follow-Up Question for the Authors**
>
> Thank you for the answers.
>
> - Do the authors plan to upload the revised paper by the communication deadline (11/26)? I believe the authors can update the paper (I saw an updated version in other assignments), and I’d like to check the revised version if possible.
> - To clarify, I suggested moving Viterbi decoding into the main methods section because the authors "propose" it in both the introduction and the experiment sections. If the authors believe it is just a method to better analyze the model, they could avoid using the term "propose" and just conduct an analysis with Viterbi decoding. That being said, I think it makes sense to "propose" it and write it in the main methods section. Finally, just to make it clear, I am not suggesting including the Viterbi decoding results in the main "table" because it is not an apples-to-apples comparison, as the authors mentioned.
> - Regarding the reranking vs. refining discussion, could the authors clarify the difference compared to the deliberation model (e.g., Ke Hu’s work)? I'm still a little cautious about calling the "refining method" a new "paradigm" (as written in the Introduction), although I now have a better understanding of the differences from prior works.
>   - Ke Hu, et al. Deliberation model based two-pass end-to-end speech recognition. ICASSP 2020.
> - I cannot agree with the argument that "It is generally reported in the literature that beam search on Transducer-based models only brings marginal improvement." While it is okay to use beam=1 in your experiment for simplicity, the improvement from beam search is usually not "marginal," and I believe it should not be used as an excuse. (Again, I am fine with the setting itself, just in case.)

---

> ### Author Response · Authors · 2024-11-25
>
> - comparison with Hu. et al
>
> We thank the reviewer pghP for bringing our attention to the paper “Deliberation model based two-pass end-to-end speech recognition” by Hu, et al. This work is indeed an improvement upon the regular "reranking" method. It first uses the fast model to generate multiple hypotheses and then, and uses a much stronger 2nd pass model, that takes in both the acoustic input, as well as the set of hypotheses, to generate the final hypothesis. In particular, in this work, the 2nd pass (slow) model has its additional encoder layers for processing speech input, bi-direction LSTMs for processing hypotheses, and is trained using a different loss function (Listen-Attend-and-Spell model instead of the RNN-T used in the 1st-pass fast model). Because the 2nd pass model has access to the whole list of k-best hypotheses, I would imagine that this approach can "invent" new hypotheses by recombining elements from different individual hypotheses. So in that aspect, this work has similarities with our work. However, Hu et al.'s 2nd pass is significantly slower, since it requires additional encoder computation, additional decoder computation, and cross-attention computation between the encoder/decoder outputs.
>
> - Regarding uploading a revision of the paper by the discussion deadline.
>
> Yes, we are working on a revision of the paper and will upload a new version by 26th. Given the tight timing, we will not be able to include additional experiment results in the revision. We will try to focus more on improving the descriptions in the paper. Some of the things we will try to accomplish in this revision are,
>
> 1. including additional discussions with related works, as suggested by reviewers
> 2. including discussions on the terminology of NAR/AR decoding to make our description more rigorous
> 3. re-organize the structure of the paper and bring the Viterbi method from the analysis to the method section.
> 4. improving the algorithm block to make the interfaces/variables more clear.
> 5. improving the discussion in Sec. 6.3, by including the points suggested by reviewer pghP23, as well as further explaining our point of "the ability to skip frames ... " by referencing Sec. 6.1.
> 6. including a discussion on why 0 duration is suppressed by HAI-T model.
> 7. fix formatting issues, 404 URL links, notation explanations for [1-8], claim the AR-HAI-T is "on par" with TDT, etc, as suggested by reviewers.
>
> While we're working on the revision, are there any particular points that reviewers would suggest that we prioritize addressing in this revision, given that we have limited time?

---

> > ### Comment · Reviewer_pghP · 2024-11-25
> > **Response to the authors' comment**
> >
> > Thank you for your effort on the revision.
> >
> > - While we're working on the revision, are there any particular points that reviewers would suggest that we prioritize addressing in this revision, given that we have limited time?
> >
> > One point I might mention is that the deliberation model is already very common in the ASR field (Hu's work has been cited 95 times in three years). There are also various extensions of it, including one to make it faster (e.g., W. Wang's work). You can easily find them by searching Google Scholar with "deliberation" or looking up papers that cite Hu's work. Therefore, when discussing the relationship to prior works, please keep this in mind and try to avoid any overclaiming.
> >
> > W. Wang et al. Deliberation of streaming RNN-transducer by non-autoregressive decoding. ICASSP, 2022.
> >
> > Thank you again for your effort.

---

### Official Review · Reviewer_jHSk · 2024-11-01

**Soundness:** 3
**Presentation:** 3
**Contribution:** 2
**Rating:** 8
**Confidence:** 4

**Summary:**

This paper proposes a simple yet effective method for training a Token-and-Duration Transducer (TDT) model by stochastically masking out the predictor. It also introduces a novel semi-autoregressive (SAR) inference mode, which first uses non-autoregressive (NAR) decoding to generate initial hypotheses and then applies autoregression to iteratively refine these hypotheses in parallel. Evaluation on multiple datasets demonstrates that the proposed method, HAI-T, achieves substantial performance gains compared to vanilla CTC in non-autoregressive mode. Furthermore, the proposed semi-autoregressive mode provides greater flexibility in balancing accuracy and speed.

**Strengths:**

* The paper introduces a straightforward approach that requires only a single-line change to the existing joint computation code in the Token-and-Duration (TDT) implementation and performs effectively in practice.
* Evaluations are conducted on multiple ASR corpora, including the AMI test, Earnings22, Gigaspeech test, Librispeech test-clean and test-other, Spgispeech test, Tedlium test, and VoxPopuli test.

**Weaknesses:**

Although the method is simple, the proposed HAI-T appears to be an incremental improvement over TDT.

**Questions:**

* In line 259, the authors should clarify what is meant by [1-8].
* The proposed semi-autoregressive inference mode generates an initial hypothesis through NAR inference and then refines the hypothesis using the predictor. How is the proposed semi-autoregressive mode different from CTC/RNN-T-based joint decoding? Could the authors expand the discussion in the related work section?
* Please provide results for the HAI-T model without duration settings in Table 2 for a better understanding across different language settings.
* Include decoding time results using Viterbi decoding in Table 3 for completeness.
* Add x and y labels in Figure 2.
* For deeper understanding, the authors could include results on RTF and emission delays for both the proposed and baseline methods.
* A realistic comparison would include the proposed semi-autoregressive inference mode against CTC/RNN-T results in joint decoding modes.
* The results appear promising. However, a detailed breakdown of the Word Error Rate (WER) improvements with semi-autoregressive mode would be beneficial. For instance, are there notable changes in substitutions, deletions, or insertions? Which error type shows the most improvement, or are they all enhanced to a similar degree? This information could provide clearer insights into the impact of semi-autoregressive decoding.
* It is difficult to understand how the HAI-T training procedure dramatically reduces the occurrence of 0-duration predictions, given that it is trained by stochastically masking out the predictor. Could the authors expand on this phenomenon in the discussion?

---

> ### Author Response · Authors · 2024-11-21
>
> We thank reviewer jHSk for the thorough review and the positive feedback.
>
> - Incremental innovation
>
> We understand that the method we propose is simple, that only a one-line change on existing code is required for training such models. We believe the simplicity of the approach is a strength of the work, since the method can be easily implemented to take advantage of the more flexible speed/accuracy tradeoff brought by our model.
>
> That being said, our SAR method is a novel innovation. Unlike alternative rescoring-based methods, which first use a fast model to generate a set of hypotheses, and then rerank them by the slower but more accurate model, our model directly uses the more accurate model to “correct” the output of the fast model, inventing new hypotheses in the process in parallel. This is a novel methodology that we believe can inspire new approaches to further the development of speech models.
>
> - Notation of [1-8]
>
> We follow the notation from [Xu et al. 2023] where [1-8] means the model supports duration predictions of 1, 2, 3, …, 7, 8. We will revise our paper to make this more clear.
>
> - Comparison with joint-CTC-RNNT decoding
>
> Can we ask the reviewer to clarify which research work the joint-CTC-RNNT decoding refers to, since it’s not a well-established term?
>
> One work we know of that is related to that is the "4D-ASR" method by Sudo et al. We’d be happy to include a discussion of the relation to that work.
>
> Or if the reviewer suggests a more generic way of joint-CTC-RNNT decoding, e.g. by first performing CTC decoding, and passing those non-blank emitting frames and tokens to the RNNT-decoder to generate the RNNT hypothesis. We feel this method might require some careful designing to work. Firstly, a jointly trained CTC-RNNT model doesn’t necessary make sure CTC and RNNT share the same alignment, and due to CTC allowing token repetitions in its text augmentation rules, there are different boundary cases we need to consider if we pass CTC outputs to RNNT decoding.
>
> - Results for the HAI-T model without duration settings in Table 2
>
> We assume the reviewer meant the HAI-T model with config without 0 duration for the German model? Sure, we can add those experiment results.
>
> - Viterbi decoding
>
> We purposefully put the Viterbi in the analysis section rather than the main results. The aim of including those results is to show that the HAI-T models can learn more diverse representations among different time indices. Using Viterbi decoding is intended as an empirical validation of this claim, which we see from consistent accuracy improvements. It’s not an apple-to-apple comparison if include Viterbi results with our main table, since the nature of the search algorithms is different.
>
> - Add x and y labels in Figure 2
>
> Thanks for noticing this issue. We will improve Figure 2 with clear and informative labels on axes.
>
> - RTF and latency metrics
>
> We agree that information of RTF and latency metrics will provide a clear picture of how HAI-T works compared to baselines. We will add those metrics.
>
> - Break down of error types
>
> This is an excellent suggestion. We will include both theoretical analysis of error patterns and also experimental results.
>
> - Discussion on why HAI-T suppresses 0 duration
>
> We will include a more detailed discussion on why duration 0 is suppressed. To explain it, let’s call encoder output E and predictor output D. The following key points are causing this phenomenon.
>
> 1. By performing stochastic masking of predictor output during training,  the model is learning to make the output distribution of joint(E_t, D_u) similar to joint(E_t, D_u * 0).
>
> 2. If duration 0 is predicted from joint(E_t, D_u), then the next step of processing would be joint(E_t, D_{u+1}), where t remains the same but u gets incremented by 1.
>
> 3. From 1, we know that model is learning to make joint(E_t, D_u) similar to joint(E_t, D_u * 0); also it’s learning to make joint(E_t, D_{u+1}) similar to joint(E_t, D_{u+1} * 0). Consequently, the model would make joint(E_t, D_u) similar to joint(E_t, D_{u+1}), since joint(E_t, D_u * 0) is the same as joint(E_t, D_{u+1} * 0), both of which has predictor masked out.
>
> 4. However, we know that joint(E_t, D_u) and joint(E_t, D_{u+1}) should predict different tokens since the text histories are different. This causes a contradiction.
>
> Therefore, the model would greatly suppress 0 duration prediction.

---

> > ### Comment · Reviewer_jHSk · 2024-11-23
> > **Response to Authors**
> >
> > >Or if the reviewer suggests a more generic way of joint-CTC-RNNT decoding, e.g. by first performing CTC decoding, and passing those non-blank emitting frames and tokens to the RNNT-decoder to generate the RNNT hypothesis. We feel this method might require some careful designing to work. Firstly, a jointly trained CTC-RNNT model doesn’t necessary make sure CTC and RNNT share the same alignment, and due to CTC allowing token repetitions in its text augmentation rules, there are different boundary cases we need to consider if we pass CTC outputs to RNNT decoding
> >
> > Yes, I was indeed suggesting a more generic approach to joint CTC-RNNT decoding. One approach could be to apply alignment regularization to enforce compatibility between the CTC and RNN-T alignments.
> >
> > >detailed discussion on why duration 0 is suppressed
> >
> > Thank you for the detailed discussion.

---

### Official Review · Reviewer_n6nQ · 2024-11-03

**Soundness:** 3
**Presentation:** 3
**Contribution:** 3
**Rating:** 8
**Confidence:** 4

**Summary:**

The paper introduces the Hybrid-Autoregressive Inference Transducer (HAI-T), an extension of the Token-and-Duration Transducer (TDT) designed to support autoregressive (AR), non-autoregressive (NAR), and semi-autoregressive (SAR) inference within a single framework. The key innovation is the use of stochastic predictor masking during training, enabling seamless switching between inference modes.

AR inference follows the TDT process, while NAR inference simplifies processing by using a zeroed tensor in place of predictor outputs for single-pass decoding. SAR inference combines the strengths of both approaches, generating an initial hypothesis with NAR and refining it with AR-like processing using shifted representations. The paper also proposes a Viterbi-based decoding method to enhance results. Experimental evaluations across English and German datasets show that HAI-T outperforms CTC in NAR mode and matches or exceeds TDT and RNN-T in AR mode, with further improvements observed using Viterbi decoding.

**Strengths:**

**Originality**: The paper introduces a simple but effective technique: masking the predictor output 50% of the time during training, which allows the joiner to handle non-autoregressive (NAR) inference. This enables the model to support multiple inference strategies—AR, NAR, and SAR. The carefully crafted NAR and SAR inference strategies leverage the TDT architecture well and show improved performance across multiple datasets.

**Quality**: The paper provides thorough experimental evaluations across various datasets in both English and German, and includes comparisons with standard models like CTC, TDT, and RNN-T. The consistent improvements in performance across these benchmarks demonstrate the robustness of the proposed approach.

**Clarity**: The authors do a great job explaining how the AR, NAR, and SAR inference modes work, making it easier for readers to follow how HAI-T switches between different inference strategies. The inclusion of a code snippet to show how the masking is implemented during training is a nice touch that makes the paper accessible to researchers at various levels of ASR expertise.

**Significance**: The introduction of NAR inference with TDT, which outperforms CTC, and the development of SAR inference, which balances the speed of NAR with the accuracy of AR, are important contributions for real-world ASR applications where both efficiency and accuracy are needed. HAI-T effectively addresses some of the limitations of existing transducer models and sets a new standard for flexible and adaptable ASR models. This work is likely to encourage more research in hybrid inference strategies within the ASR field.

**Weaknesses:**

1. **Incremental Architectural Innovation**: While HAI-T's hybrid approach to inference is creative, it builds on the existing Token-and-Duration Transducer (TDT) model. The primary novelty lies in its training strategy and inference flexibility rather than any fundamental architectural innovations. This could make the contribution seem incremental.

2. **Lack of Detailed Training/Inference Specifications**: The paper omits crucial details about the training setup, such as the type and number of GPUs used, learning rates, schedules, and batch sizes. Additionally, the reported inference times lack confidence intervals and clarity on whether they were computed using a CPU or GPU. These omissions limit reproducibility and make it difficult to gauge the true computational demands of the model.

3. **Dependency on Pretrained Models**: The reliance on pretrained models for initializing the encoder, which are already robust and highly optimized, raises questions about the origin of the reported performance gains. This reliance could make it challenging to separate the contribution of HAI-T's unique elements from the advantages conferred by the pretrained encoder. Furthermore, the use of an English checkpoint for initializing the German model might influence the results, yet the paper does not address this potential impact.

4. **Clarity on Limitations**: The paper would benefit from a clearer discussion of the potential limitations of HAI-T. For instance, it does not explore how the model performs under challenging conditions, such as noisy or highly varied audio inputs, or whether it would require significant adjustments to maintain performance in such scenarios.

**Questions:**

1. Why was the Viterbi-based decoding added as an after-thought rather than in the main experiment? By how much does it increase the inference times?

1. Could you provide more details on the training setup? The link to hyperparameters in footnote 3 gives a 404-not found error.

2. How were the inference times computed, and provide the confidence intervals?

3. Did you try training the HAI-T from scratch? Especially since you have access to a large English dataset, training from scratch should be feasible.

4. What are the potential impacts of initializing the German model with an English encoder checkpoint?

5. When applied to data with high variability or noise, how does the model handle such conditions compared to existing approaches?

6. Table 1 and 3 exceeds the margin, please ensure it fits within the margins.

7. The algorithms could use some refinement to avoid ambiguity in mathematical operations and naming conventions. For example `shifted_hyp` instead of `shifted-hyp`

---

> ### Author Response · Authors · 2024-11-21
>
> We thank reviewer n6nQ for the thorough and positive review of the paper.
>
> - Incremental innovation
>
> We understand that the method we propose is simple, and that only a one-line change on existing code is required for training such models. We believe the simplicity of the approach is a strength of the work, allowing the method to be easily implemented to take advantage of the more flexible speed/accuracy tradeoff brought by our model.
>
> That being said, our SAR method is a novel innovation. Unlike alternative rescoring-based methods, which first use a fast model to generate a set of hypotheses, and then rerank them by the slower but more accurate model, our model directly uses the more accurate model to “correct” the output of the fast model, inventing new hypotheses in the process while running in parallel. This is a novel methodology that we believe can inspire new approaches to further the development of speech/language models.
>
> - Lack of detailed training/inference specs
>
> We apologize that the code link we provided in the paper links to a 404 error. It seems the official NeMo repository has been updated since our submission and the config file has been moved to https://github.com/NVIDIA/NeMo/blob/main/examples/asr/conf/fastconformer/fast-conformer_transducer_bpe.yaml. We will make sure to include the latest link in the revised version of the paper.
>
> For inference time, we measure the wall time. We take the reviewer’s suggestion and plan to include more details, including machine/GPU specs for inference in our evaluations.
>
> - On the use of initialization from existing checkpoints
>
> We use initialization to speed up the experiments. At the beginning of the development, we also ran experiments training from scratch and noticed the same performance/speed patterns with those models, and the only difference is that many more training steps are required for models to converge.
>
> This faster convergence is also seen when we initialize the German model with an English checkpoint. While those languages are different, it’s reasonable to assume at least in the early layers where the network mostly performs feature extraction, an English model checkpoint can still benefit German model training. In practice, the converge speed gain isn’t as great as English models, but still faster than from scratch.
>
> - On the limitation of the model in noisy/hard scenarios
>
> The reported results on librispeech test-clean and test-other touched on this aspect, as test-other is a noisier version of the librispeech dataset, which corresponds to the slightly degraded performance observed among all models. And from the numbers, there are no clear patterns in terms of which model is significantly more noise-robust than others, and the existence of noise impacts all models to similar degrees.
>
> - On Viterbi decoding as an after-thought
>
> Yes, we purposefully put the Viterbi in the analysis section rather than the main results. The aim of including those results is to show that the HAI-T models can learn more diverse representations among different time indices. Using Viterbi decoding is intended as an empirical validation of this claim, which we see from consistent accuracy improvements.  It’s not an apple-to-apple comparison if include Viterbi results with our main table, since the nature of the search algorithms is different.
>
> - Table 1 and 3 are too wide.
>
> Thanks for noticing this. We will fix it.
>
> - Notations in Algorithm box.
>
> We will take the advice and improve the clarity of the algorithm.

---

> > ### Comment · Reviewer_n6nQ · 2024-11-22
> > **Response to rebuttal**
> >
> > Thanks for the clarifications.
> >
> > >In practice, the converge speed gain isn’t as great as English models, but still faster than from scratch.
> >
> > If possible, I would like to see a comparison between the German model trained from scratch and initialized with English checkpoint.
> >
> > > Yes, we purposefully put the Viterbi in the analysis section rather than the main results.
> >
> > Well, I still think the Viterbi decoding should be part of the main proposed method section since it is one of the three claimed contributions in the introduction.

---

> > > ### Author Response · Authors · 2024-11-22
> > >
> > > Thanks for the additional feedback.
> > >
> > > We will work on including the from-scratch German results in the paper, as well as moving the Viterbi from the analysis to the main method section.

---

### Official Review · Reviewer_rxjr · 2024-11-03

**Soundness:** 3
**Presentation:** 3
**Contribution:** 2
**Rating:** 6
**Confidence:** 4

**Summary:**

This paper proposes a training method to improve the performance of Token-and-Duration Transducer (TDT, Xu et al., 2023), which enables TDT to perform decoding in several different modes that improve efficiency without sacrificing performance much.

The main contributions are as follows
1. Introduce “transcript masking” as a regularization method for TDT training, which improves the regular auto-regressive decoding performance
2. Decode with “empty transcript” (ie feed entirely masked transcript to the predictor module) to greatly reduce the runtime for AR decoding. (This is referred to as non-autoregressive (NAR) decoding in the paper, but I do not agree it should be called NAR. See details in later sections)
3. A hybrid decoding approach to improve the hypothesis proposed by empty transcript decoding. In particular, it uses that hypothesis as input the predictor module, gets predictor embeddings for each text position “in parallel”, and the take the argmax from each position in parallel.

**Strengths:**

1. Masking target is an effective approach verified in several other models (TDS for CTC, MaskPredict for NAR translation), which can serve as a method of regularization or a way to build non-autoregressive model. Applying to TDT is a clever idea which serves both purposes.
2. Empirical results on large scale study are convincing in terms of the effectiveness of improving NAR and the more efficient decoding methods (NAR, SAR) proposed in the paper based on TDT
3. The paper also presents a good set of ablation studies (stateless / no zero duration / duration range / duration distribution analysis / argmax token analysis) to provide readers more insights on why it works and how the model behaves.

**Weaknesses:**

1. Technically it is not correct to call the method non-autoregressive. Despite that the predictor does not take the predicted token as input, it still depends on the duration predicted from the previous step to determine what enc[t] to compute argmax on (line 7 of Xu et al. (2023)). Hence, they cannot be fully parallel and should be still considered autoregressive. I acknowledge that runtime wise it is almost identical to NAR because argmax can be precomputed for all (t, u), but this is still wrong in terms of the nomenclature.
2. Missing discussion of limitations. For example, the refinement step cannot fix the error if the NAR hypothesis is shorter or longer than the ground truth.
3. Missing comparison with similar or related methods. For example, how does it compare with generating top-K hypothesis with HAI-T NAR or CTC, and then rescore by an AR model
4. The argument of HAI-T being superior than RNN-T and TDT is too strong. On individual datasets it loses in quite a few cases (e22, giga, clean, other, spgi, vox). The authors should just claim on-par.

**Questions:**

* (Line 254) How important is it to initialize HAI-T with TDT? Are the baseline models’ encoder initialized from a PT model? What’s the performance drop for HAI-T if it is not initialized from TDT?
* Is the stateless predictor merely an embedding table?
* Has the authors compared NAR with NAR-Viterbi? How are the duration posterior and token posterior combined (ie is a weight introduce to rebalance token and duration posterior)?

---

> ### Author Response · Authors · 2024-11-21
>
> We thank reviewer rxjr for the thorough review of our paper.
>
> - On the terminology of NAR
>
> We appreciate the reviewer rxjr's careful analysis of our NAR terminology. Our decoding process does maintain some sequential dependencies through duration-based frame selection, so strictly speaking, the decoding procedure is not NAR.
>
> We would be happy to revise our writing and emphasize that when we call the model NAR, it specifically regards the model’s neural network computation. Note, that the AR/NAR categorization for models usually refers to the neural network computations but not the complete procedure. As a point of comparison, even CTC models - widely accepted as non-autoregressive (e.g many papers from https://scholar.google.com/scholar?q=CTC+non+autoregressive), have implicit sequential dependencies in their decoding (e.g., blank token handling and repeated token resolution would depend on prior predicted tokens), yet are considered NAR due to their parallel neural computations.
>
> - On the limitation of SAR methods
>
> We agree that there are limitations in terms of the errors SAR methods can fix. We had planned a section for that but was later removed due to space constraints. It’s quite interesting that despite limitations at the subword level, SAR can correct all three types of errors (deletion/substitution and insertion) at the word level.
>
> 1. From the subword level, the SAR methods can correct substitution errors and insertion errors (by replacing an original token with blank). It can't correct deletion errors, since as the reviewer correctly points out, the SAR output can't be longer than the original hypothesis.
>
> 2. From the word level though, SAR can actually correct all three types of errors, including deletion errors (where hypothesis is shorter than the reference). E.g. if the reference text is “black board”, and the initial hypothesis is “blackboard”. This results in a substitution error and a deletion error; it’s possible for SAR to replace the subword “board” with “_board”, thus reducing both the substitution error and deletion error.
>
> Since this issue was brought up by reviewers, we’d be happy to include the discussion on this topic as well as stats on the distribution of those error corrections in our revision of the paper.
>
> - On comparison with top-k + rescoring methods
>
> We point out that SAR is orthogonal to methods like k-best rescoring. With k-best rescoring, it uses a stronger model to rerank and pick the best among existing hypotheses, and this process is usually quite computationally extensive; with our SAR approach, it can invent new hypotheses based on old ones. As we’ve shown in the paper, SAR can be done extremely fast.
>
> It is possible to combine k-best restoring methods with SAR techniques, which we set as our future work.
>
> - On the claim of HAI-T being on par with TDT
>
> We thank the reviewer and take the suggestion to claim that the models are on-par in terms of performance.
>
> - On the importance of initialization from existing checkpoints
>
> We use initialization to speed up the experiments, and yes, all experiments reported in the paper including the baselines are trained with checkpoint initialization, so they're all fair comparisons. At the beginning of the development, we also ran experiments training from scratch and we noticed the same performance/speed patterns with those models, and the only difference is much more training steps are required for models to converge.
>
> - On the implementation of the stateless predictor
>
> We use the official stateless predictor implementation at https://github.com/NVIDIA/NeMo/blob/main/nemo/collections/asr/parts/submodules/stateless_net.py. And yes, the stateless predictor stores a set of embedding tables per location; when it takes input, it does an embedding look-up, and outputs the concatenated embeddings as its output.
>
> - On NAR VS NAR-Viterbi
>
> There is no extra hyperparameter we use to combine the token and duration probabilities. They are simply added in the log space with 1:1 weight.
>
> NAR results are shown in Table 1, and in Table 3, we didn’t repeat those results of NAR due to space constraints, since readers can still compare them across tables. If space permits, we’d be happy to duplicate rows in Table 3 for better comparison. That being said, we include the Viterbi results mainly to demonstrate that the HAI-T model can learn diverse representations dispersed among different time stamps. The reported improved accuracy of the Viterbi method is to prove that the representation learned by HAI-T models is indeed better. We understand that by including the Viterbi results, a comparison between that and the NAR/AR method is inevitable but those aren't really fair comparisons due to the different nature of those inference algorithms.

---

> > ### Comment · Reviewer_rxjr · 2024-11-26
> > **Response to author comment**
> >
> > I thank the authors for the thorough responses.
> >
> > Please include the changes mentioned in the response (clarify sequential dependency, discussion on limitation, error distribution before/after SAR refinement)).
> >
> > I agree that k-best is orthogonal and can be combined with SAR. However, I don't fully agree that rescoring needs to be done with a much stronger model. Similar to what's shown in the paper, using the same model but with AR-based inference may lead to more accurate posterior estimation compared to NAR-based inference. So my suggestion was to try it with the same model as done here. That said, while this is related, I agree this can be left for future work.
> >
> > I will keep my current rating.

---

### Author Response · Authors · 2024-11-27
**Paper revision submitted**

We've uploaded a revised version of the paper addressing most of the concerns raised by reviewers. Here are a short list of changes:

1. we added more discussions about related work suggested by reviewers. In particular, we emphasized a strength and novelty of this work when comparing it to other fast/slow methods (e.g. deliberation model, reranking model, etc) where our "slow" model only induces marginal parameters/computation time on top of the "fast" model, while with those other methods, the "slow" model typical include more encoder and decoder components and add considerable processing time.

2. we added a discussion on the use of the terminology of "non-autoregressive models" in our writing to make our claims more rigorous.

3. we reorganized the paper to move the Viterbi part from the analysis to the main method.

4. we improved our Algorithm blocks, by 1. including the algorithms for AR and NAR inference for the HAI-T model for completeness. 2. adding shape information of different tensors referenced in the Algorithm, to improve their clarity.

5. we added a discussion on the limitations of the SAR method and presented stats on the breakdown of error types corrected by SAR methods. We also included some real examples of the corrections in the Appendix section.

6. we added a discussion on why 0 duration is mitigated by the HAI-T model, in the Appendix.

7. we fixed minor issues raised by reviewers (URLs, notation of Algorithm, the claim that HAI-T is on-par with TDT/RNNT, explanation of duration setting [1-8], etc).

Due to time constraints, we are not able to include more baseline experiments. We are dedicated to running them and including them in the final version if the paper is accepted. We thank all reviewers again for their thorough review of our paper and valuable feedbacks.

---

> ### Comment · Reviewer_pghP · 2024-11-28
> **Response to the revised paper from reviewer pghP**
>
> Thank you for the update.
>
> - The authors still seem to be sticking with the statement of "present a novel paradigm" even after including many prior works in the related work section, which I am really hesitant to agree with. As I have repeatedly mentioned, the notions of slow-fast decoding, deliberation, and semi-autoregressive methods are all known concepts, and I don’t see the proposal as a novel "paradigm" nor a novel "approach," although I acknowledge it is a novel "method." I’d recommend the authors change "paradigm" to "method" in line 50. I also recommend fixing line 528 to avoid the wording of "approach," which also implies the proposal is more than a method.
>
> - I didn’t recommend including the Viterbi result in Table 1. (As I mentioned in my prior comment, what I suggested was just moving the method explanation into the main method section.) However, given the suggestion was made by another reviewer, I am fine with keeping the current form.
>
> - Given that the paper has been updated, I have raised the soundness and presentation scores as well as overall rating. I still rate the overall score as marginally below acceptance, given that I think the novelty is incremental and not all comments have been addressed. My rating may be too severe, and I’d like to follow the decision by a meta-reviewer in this regard.

---

### Meta-Review · Area_Chair_9HvG · 2024-12-20

**Metareview:**

In this paper, the authors proposed Hybrid-Autoregressive Inference Transducers (HAI-T) which further improves the Token-and-Duration Transducer (TDT) model. The key idea is to randomly mask predict network outputs during training. In this way, a single HAI-T model can be evaluated with three modes: autoregressive (AR), non autoregressive (NAR), and semi-autoregressive (SAR). The implementation is extremely simple, only one line change of the original TDT code. The authors have conducted extensive experiments on multiple ASR test sets,  including AMI, Earnings22, Gigaspeech, Librispeech, Spgispeech, Tedlium, and VoxPopuli. The performance gain on all these tasks makes the contribution more convincing.

The proposed method demonstrates simplicity and effectiveness. A crucial aspect of the review process is assessing the novelty of the proposed method. The author has claimed that they introduced a novel “paradigm,” which seems to be an overstatement. I concur with Reviewer pghP that this resembles a novel method with only a single line code change from TDT training. The authors are advised to temper their claim in the final version. There is also discussion regarding its distinction from 2nd pass rescoring. I align more with the authors, as traditional 2nd pass rescoring entails significantly higher costs, whereas the proposed method is lightweight. During the rebuttal process, the authors have made considerable efforts to enhance the paper's quality by adding discussions on related work, clarifying the algorithm description, and acknowledging the limitations of SAR, among other improvements.

Overall, the paper has its own values to audience.

In summary, the strength of this paper is that it presents a simple but effective method to enable AR/SAR/NAR architecture in a single model, significantly boosting the performance. The weakness of the paper is that the authors somehow overclaimed the method’s significance.

**Additional Comments On Reviewer Discussion:**

During rebuttal, the authors have clearly answered all the questions from reviewers. The authors has a very good summary of all the revisions in one thread of the reviewers/authors discussion. The initial scores of this paper are split with 3, 6, 8, 8. After the rebuttal, the final scores are 5, 6, 8, and 8. Note that the major concerns came from Reviewer pghP. As the authors have answered most of his/her questions, the score was boosted from 3 to 5 by Reviewer pghP. However, I agree with Reviewer pghP that the paper has over claimed this method as novel paradigm. Hope the authors can tone down their claim as a novel method in the final version.

---

### Decision · Program_Chairs · 2025-01-22

Accept (Poster)